# Photonic crystal cavities from hexagonal boron nitride

Sejeong Kim[1], Johannes E. Fröch[1], Joe Christian[2], Marcus Straw[2], James Bishop[1], Daniel Totonjian [1],
Kenji Watanabe [3], Takashi Taniguchi[3], Milos Toth [1] & Igor Aharonovich[1]

Development of scalable quantum photonic technologies requires on-chip integration of photonic components. Recently, hexagonal boron nitride (hBN) has emerged as a promising platform, following reports of hyperbolic phonon-polaritons and optically stable, ultra-bright quantum emitters. However, exploitation of hBN in scalable, on-chip nanophotonic circuits and cavity quantum electrodynamics (QED) experiments requires robust techniques for the fabrication of high-quality optical resonators. In this letter, we design and engineer suspended photonic crystal cavities from hBN and demonstrate quality ($Q$) factors in excess of 2000. Subsequently, we show deterministic, iterative tuning of individual cavities by direct-write EBIE without significant degradation of the $Q$-factor. The demonstration of tunable cavities made from hBN is an unprecedented advance in nanophotonics based on van der Waals materials. Our results and hBN processing methods open up promising avenues for solid-state systems with applications in integrated quantum photonics, polaritonics and cavity QED experiments.

[1] Faculty of Science, Institute of Biomedical Materials and Devices (IBMD), University of Technology Sydney, Ultimo, NSW 2007, Australia. [2] Thermo Fisher Scientific, 5350 NE Dawson Creek Drive, Hillsboro, OR 97214-5793, USA. [3] National Institute for Materials Science, 1-1 Namiki Tsukuba, Ibaraki 305-0044, Japan. These authors contributed equally: Sejeong Kim, Johannes E. Fröch. Correspondence and requests for materials should be addressed to S.K. (email: Sejeong.Kim-1@uts.edu.au) or to M.T. (email: milos.toth@uts.edu.au) or to I.A. (email: igor.aharonovich@uts.edu.au)

Controlling and manipulating light at the nanoscale is important for a vast variety of applications that include sensing, quantum information processing, secure communications and cavity QED experiments[1–11]. Key components for most of these applications include optical resonators such as photonic crystal cavities, and non-classical light sources that can emit single photons on demand such as color centers in solids[12], defects in carbon nanotubes (CNTs)[13], single molecules[14] and quantum dots[15]. Remarkable progress has been achieved over recent years to realize on-chip integrated quantum photonic circuits that employ various combinations of these systems. For example, in a monolithic approach where nanophotonic elements and a quantum light source are embedded in the same material, coupled systems have been implemented using diamond[5,11], rare earth crystals[4], and gallium arsenide[16]. Alternatively, hybrid systems, where an external source is positioned in close proximity to the photonic element made from a foreign material, have been assembled from a broad range of materials such as CNTs[17] and InAsP quantum dots (QDs)[18,19] integrated with silicon nitride components.

In recent years, layered van der Waals materials have emerged as promising hosts of ultra-bright quantum emitters[20]. Integration of these light sources with dielectric and metallic waveguides has been achieved by placing flakes of the van der Waals hosts on top of the waveguides[21–23]. However, in such a hybrid approach, the emitter couples only to the evanescent field of the cavity mode. Hence, spatial matching between the emitter and the electric field maximum is limited, and scattering losses are increased. A monolithic system, in which the photonic resonator hosts the quantum emitter is required for ideal on-chip devices.

Here, we design and fabricate optical cavities from hBN – a wide bandgap, hyperbolic van der Waals material[24] that has recently attracted considerable attention as a promising host of ultra-bright, room-temperature quantum emitters[20,25–29]. In addition, hBN is naturally hyperbolic and exhibits volume-confined phonon polariton modes[30] that open up opportunities to study light–matter interaction in the deep subwavelength regime and advance infrared and terahertz nanophotonics[30,31]. It is therefore important to engineer nanostructures that enable control over a broad spectral range that accommodates both visible on-chip nanophotonics and integrated polaritonics. Finally, the layered nature of hBN allows flakes to be easily relocated and combined with other platforms. This, together with a high degree of inherent chemical inertness, makes hBN resilient to a broad range of environments such as liquids, and hence applications in sensing where the relatively low refractive index of hBN is beneficial as it leads to large evanescent fields that enhance sensing efficiency.

## Results

**Fabrication hBN photonic crystals**. As a first step toward applications, we demonstrate nanofabrication of two-dimensional (2D) and one-dimensional (1D) photonic crystal cavities (PCCs) with optical $Q$-factors of up to 2100. Such fabrication protocols are needed because, in contrast to mature semiconductors, techniques for making high Q-factor optical devices from van der Waals layered materials that operate in the visible wavelength range are not yet fully developed. For example, it was unknown whether 2D stacked layers are sufficiently robust to endure nanofabrication processes without being destroyed. Also, under-cutting techniques that are used to achieve suspended structures and bulk angle etch processes have not been applied successfully to these materials. This is particularly significant because of the small refractive index of hBN of $\sim 1.8$ (at $\lambda = 600$ nm) makes it hard to achieve a high refractive index contrast that is needed for efficient light confinement in the visible spectral range. In this work, we resolve these challenges by demonstrating the fabrication and iterative editing/tuning of suspended photonic cavities from the van der Waals material hBN using a combination of hBN exfoliation onto a trenched substrate, reactive ion etching (RIE) and single-step, direct-write electron beam induced chemical etching[32].

As a first step toward the nanofabrication of photonic cavities, we prepared hBN flakes by scotch tape exfoliation from bulk crystals, which were synthesized in a high pressure–high temperature process that yields carbon and oxygen impurity concentrations below $10^{18}$ cm$^{-3}$ [33]. Examples of exfoliated hBN films on a bulk substrate, as shown in Fig. 1a, reveal a sufficient lateral size for fabrication of devices on the micrometer scale, as films appear smooth over several microns with low amounts of grain boundaries, while the different thicknesses of hBN stacks are apparent by their different colors. Due to the low refractive index of hBN, substrates cause severe optical losses. We therefore employed a substrate with pre-patterned trenches so that optical cavities can be fabricated in the suspended regions, as schematically depicted for the case of a 2D photonic crystal in Fig. 1b. Figure 1c is a scanning electron microscope (SEM) image of the suspended hBN showing the layered nature of hBN. Layers are held together by van der Waals forces and each 2D monolayer consists of alternating boron and nitrogen atoms arranged in a hexagonal pattern. The thicknesses of hBN flakes vary from a monolayer to a few hundred nanometer and these flakes appear smooth and flat, as is shown for several examples in the Supplementary Note 1. From exfoliated hBN flakes, only those with a thickness in the range of 200–500 nm were considered for fabrication. The lower limit was chosen to provide a sufficient $Q$-factor, as thinner films will not confine light sufficiently, whereas the upper limit is selected to allow for only a single guided mode in the out-of-plane direction, as generally preferred for photonic cavities.

The first resonators that we show are 2D PCCs fabricated using a nanofabrication procedure that combines RIE with EBIE. Briefly, a tungsten layer that serves as a hard mask is deposited on top of hBN, followed by a thin layer of e-beam resist (polymethyl methacrylate (PMMA)). Using electron beam lithography, the 2D PCC pattern is written in the PMMA film followed by an RIE step in SF$_6$ gas, which transfers the 2D PCC pattern into the tungsten mask. The underlying hBN is then etched by material-selective EBIE using water vapor as a precursor gas[34], which results in nearly-straight sidewalls. Moreover, EBIE is a chemical process which does not rely on physical removal of atoms through knock-on processes that commonly occur in RIE and FIB techniques and cause substantial damage in the host crystal[32]. As a result, no postprocessing (via annealing or wet-chemical treatment) is necessary after EBIE in order to observe and subsequently tune cavity modes, as is shown below. The EBIE process does not involve heavy ions (e.g., Ga) hence re-sputtering of ions and re-deposition of the etched material are absent. The processing steps and a comparison with other etching methods are explained further in the Supplementary Notes 2 and 3, respectively.

Figure 1d shows a side view, false-color SEM image of the 2D PCC fabricated in a suspended flake of hBN with nine holes missing in the center (L9 cavity). A top-view SEM image of the same cavity is shown in Fig. 1e. The geometries of the PCCs presented here were designed to have resonances in the visible spectral range[35], where hBN quantum emitters are typically observed. To achieve the desired resonances, we used a lattice constant ($a$) in the range of 240–300 nm, with an air hole radius in the photonic mirror region of $0.33a$ and two air holes at the end of the line defects with $0.22a$, shifted outwards by $0.22a$.

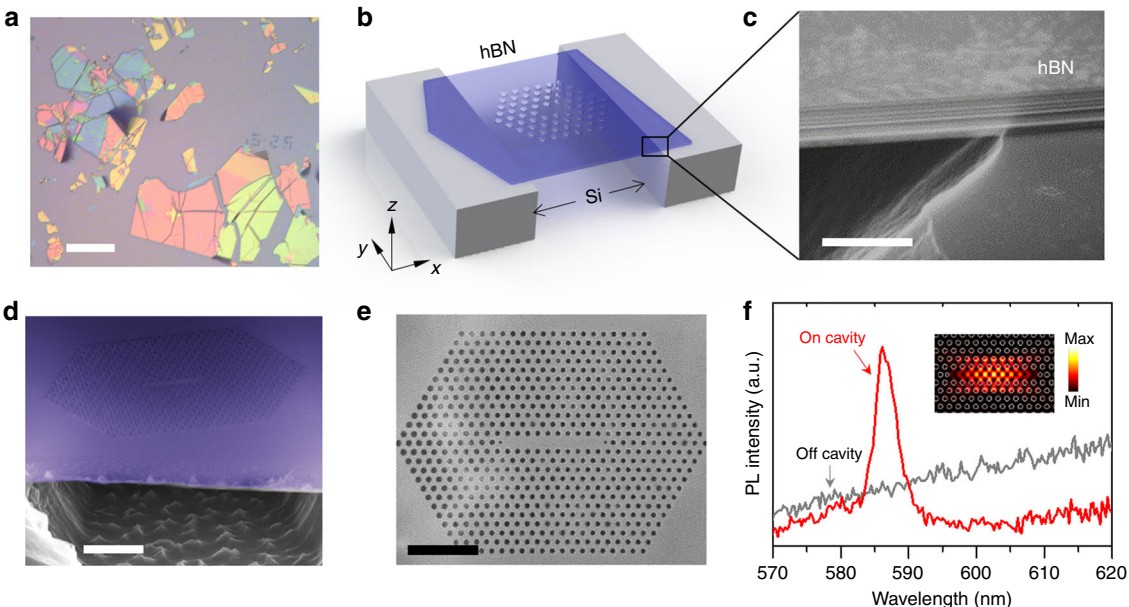

**Fig. 1** Free-standing hexagonal boron nitride 2D photonic crystal cavities. **a** Optical microscope image of exfoliated hBN crystals on a silicon substrate. The scale bar corresponds to 50 μm. **b** Schematic of a free-standing hBN cavity on a trenched silicon substrate. **c** SEM image of the hBN showing the layered structure. The scale bar corresponds to 500 nm. **d** false color SEM image (45°) of a free-standing hBN photonic crystal cavity fabricated using a combination of RIE and EBIE. The scale bar corresponds to 2 μm. **e** Top view of a 2D photonic crystal cavity. The scale bar corresponds to 2 μm. **f** Photoluminescence spectra with a laser exciting the cavity mode (red) compared with an off-cavity excitation (gray). The inset depicts the electric field intensity profile of the fundamental mode for the cavity calculated using 3D FDTD

The fabricated cavities were analyzed optically using a confocal microscope setup at room temperature. A broadband hBN background emission is excited by a 532 nm continuous wave laser, and an objective lens with a numerical aperture of 0.9 is used for excitation and collection. A comparison of room temperature photoluminescence (PL) spectra measured by excitation of the cavity center with line defects (red) compared to excitation of an adjacent periodically patterned PCC area (gray) is shown in Fig. 1f. Excitation of the cavity yields an optical mode at 586.6 nm, whilst only a broad PL emission is observed when the laser spot is off the cavity. The electric field intensity profile of the measured mode is depicted in the inset, calculated using the 3D finite difference time domain (FDTD) method. By using a Lorentzian fit, a $Q$-factor of 160 is obtained. The observed $Q$-factor is relatively low, yet there are no other studies on 2D PCCs with similar or lower refractive index materials in the visible wavelength range. We note that, in practice, it is very challenging to open a photonic bandgap using a low refractive index material and a 2D cavity geometry.

**1D Photonic crystal nanobeam.** To achieve high $Q$ cavities from low refractive index materials, it is more favorable to fabricate 1D ladder-type PCCs. Figure 2a shows representative SEM images of a 1D ladder PCC (beam width $w = 750$ nm) fabricated using the same combination of RIE and EBIE that was used to fabricate the cavity in Fig. 1. The ladder contains 25 uniform-sized, rectangular air holes ($h_x = 150$ nm, $h_y = 550$ nm) with a lattice constant $a$ of 250 nm (shown in Fig. 2b). The lattice constants of 19 air holes are modulated to form a photonic well at the center of the ladder structure with decreased lattice constants in the cavity region compared with the mirror region.

Detailed information on the cavity design and the formation of the depicted modes from the photonic band is given in Supplementary Note 4. The designs must accommodate a range of mode wavelengths in order to enable applications such as sensing and integrated photonic devices. This is particularly

important in the case of hBN since quantum emitters in this material possess a relatively wide range of emission wavelengths[36], thus necessitating the fabrication of 1D PCCs with modes spanning ~576–762 nm. Figure 2c shows room temperature PL spectra acquired from three 1D ladder PCCs with lattice constants of 220 nm (black), 250 nm (red), and 280 nm (blue), and resonances that span 600–750 nm, and thus cover most of this range. The three cavities were fabricated in the same hBN flake with a thickness of 280 nm. The 1D ladder cavity design has three confined resonant modes from the lowest energy photonic band as is observed in the PL spectrum obtained from the cavity with $a = 220$ nm (Fig. 2c). These three modes are in good agreement with a spectrum obtained from FDTD simulation (see Supplementary Note 4). The zeroth order mode, also called the fundamental mode, from the first dielectric band (F0) is observed at 612.8 nm, while first order (F1) and second order (F2) modes are observed at 633.9 and 655.5 nm, respectively. Here, "F" designates that the modes originate from the "First" dielectric band, and the numbers 0, 1, and 2 represent the number of nodes along the $x$-axis. All of these modes are both TE modes and the dielectric modes, for which the electric field maximum is in the higher dielectric region (i.e., within the suspended hBN beams). Cavities with larger lattice constants have optical modes at longer wavelengths. For example, the measured F0 mode is at 678.4 nm for $a = 250$ and 756.1 nm for $a = 280$ nm (Fig. 2c). Higher energy modes, in this case the zeroth order mode from the second dielectric band (S0), which are of interest for sensing applications due to their extended evanescent fields[37], are also observed for cavities with $a = 250$ and 280 nm. The electric field profiles of all observed modes, determined by 3D FDTD simulations, are depicted in Fig. 2d.

We fabricated and analyzed a number of cavities with a range of lattice constants. The $Q$-factor as a function of wavelength for both the first dielectric (F0, red circle) and the second dielectric (S0, black triangle) modes are summarized in Fig. 2e, including the cavity modes shown in Fig. 2c. Here among F0, F1, and F2, we

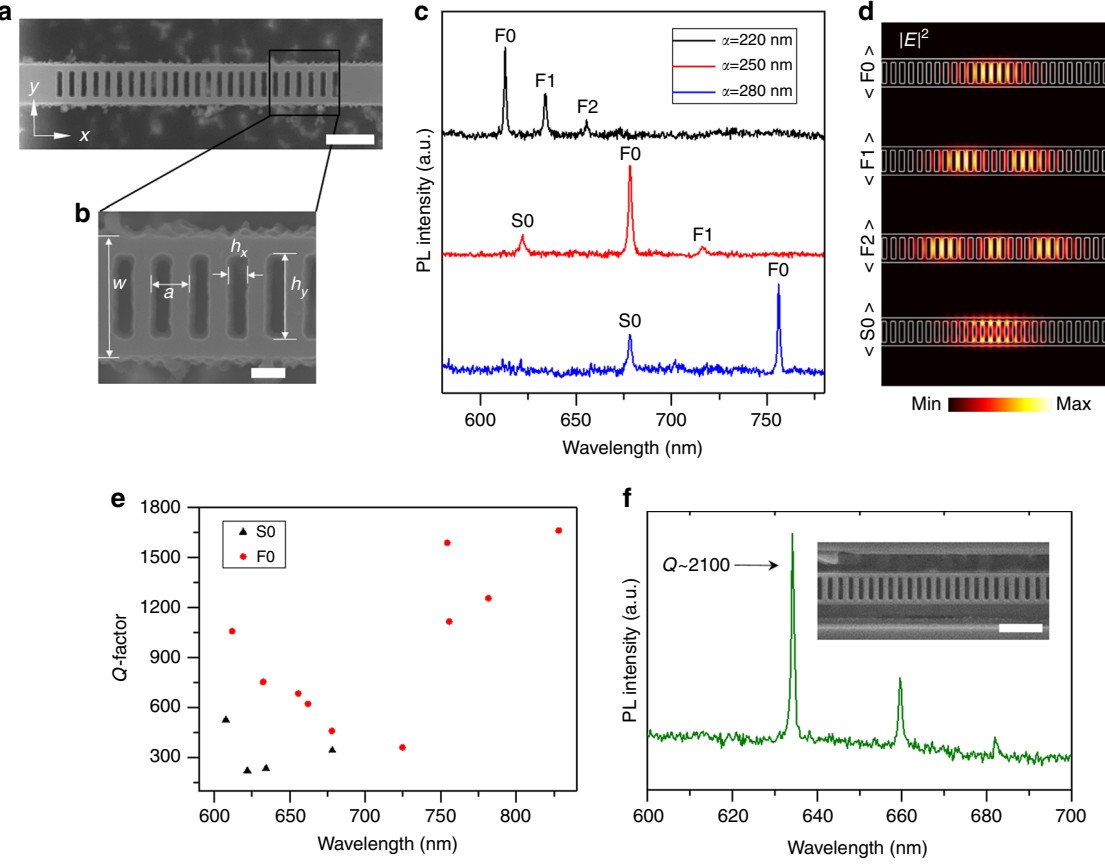

**Fig. 2** Optical analysis of one dimensional (1D) hBN photonic crystal cavities. **a** SEM image of a 1D ladder PCC with 25 rectangular air holes. The scale bar is 1 μm. **b** magnified view showing the geometrical parameters; width ($w$), lattice constant ($a$), air hole width ($h_x$) and air hole height ($h_y$). The cavity was fabricated using a combination of RIE and EBIE. The scale bar corresponds to 200 nm. **c** Photoluminescence spectra of different 1D ladder PCCs in the same hBN crystal with varying lattice constants of 220 nm (black), 250 nm (red), and 280 nm (blue), respectively. F0, F1, and F2 mark the position of the zeroth, first, and second order mode from the first dielectric band. S0 marks the position of the zeroth order from the second dielectric band mode. **d** 3D FDTD simulation result showing field profiles of the measured optical modes. **e** Experimentally obtained $Q$-factors of various 1D PCCs fabricated from hBN. **f** PL spectrum of a 1D cavity fabricated by focused ion beam milling, showing a high $Q$ (~ 2100) mode in the visible spectral range. The inset is an SEM image of the cavity and the scale bar corresponds to 1 μm

only included F0 because modes with a lower number of nodes have higher $Q$-factors. The highest $Q$-factor measured from these cavities is 1700. Theoretical $Q$ for a 1D photonic crystal cavity obtained using the 3D FDTD method for the measured parameters $a = 250$ nm, $w = 750$ nm, $h_x = 150$ nm and $h_y = 550$ nm corresponds to 9000 with a resonant wavelength of 662.8 nm. The experimental $Q$ is lower than the theoretical value mainly due to the roughness of the sidewalls seen in Fig. 2a. We also note that higher $Q$-factors are expected from cavities with resonances at longer wavelengths because scattering caused by surface roughness is reduced. It is important to note that this is the first report of such experimental $Q$ values in a resonator made entirely from a layered van der Waals material.

To further investigate potential techniques for the fabrication of cavities from suspended hBN, we employed a focused ion beam. While FIB is an attractive option for direct milling of some materials, it causes damage within the crystal that must be recovered by annealing. However, the effectiveness of annealing treatments varies substantially for different materials. For example, diamond cavities fabricated using a FIB have very low $Q$ (<800)[38] while yttrium orthosilicate (YSO) crystals show higher $Q$-factors of several thousands[4]. Figure 2f shows the PL spectrum of a cavity that was fabricated by FIB milling, and annealed (900° C, Vacuum, 2 h) to recover FIB-induced damage. Additional details on fabrication of cavities by FIB milling is given in the

Methods section. Modes are observed at 634.2 nm (F0), 659.6 nm (F1), and 682.0 nm (F2), with a $Q$-factor as high as 2100 for the F0 mode. We note that no modes were observed immediately after FIB fabrication, and the annealing step is required to remove residual crystal damage. The inset is an SEM image of the cavity. A detailed comparison of cavities fabricated by both methods is given in the Supplementary Note 5 and 6, which shows that for EBIE-fabricated cavities sidewalls within the air holes appear more vertical but rougher, while FIB-fabricated cavities show smooth sidewalls that are slanted. However, Raman analysis shows no evidence of damage directly after fabrication by EBIE, whilst residual damage is evident after FIB milling even after the annealing treatment that is required to observe optical modes. Hence, at present, the limitations and variations in $Q$-factor of cavities fabricated by EBIE and FIB are determined by surface roughness and crystal damage, respectively. The fact that EBIE processing does not require annealing is highly favorable for subsequent editing and iterative tuning of the cavities, as is discussed below. Moreover, this method can be potentially expanded to be totally mask-free to directly fabricate devices as shown in Supplementary Note 7.

The obtained cavities from hBN are on par, yet lower than other bulk dielectric semiconductors that have much higher refractive indices in the visible wavelength range, as is summarized in the Supplementary Note 8. Here, we focus on

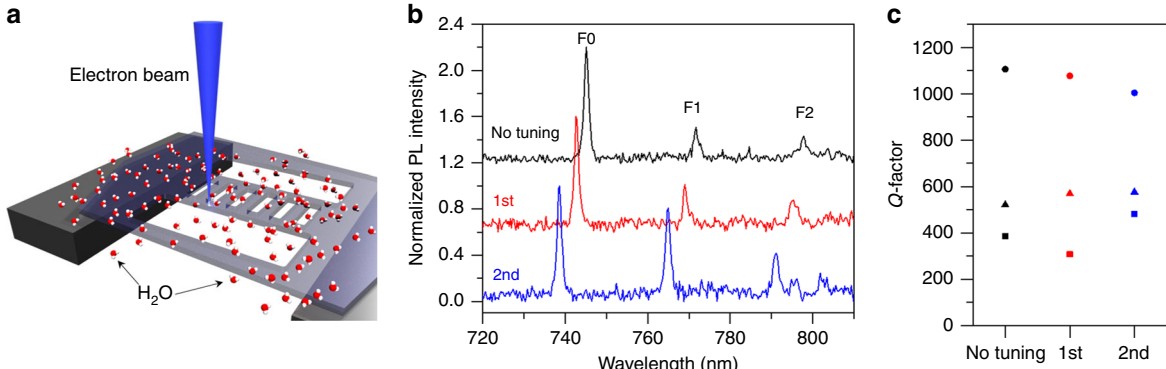

**Fig. 3** Tuning of a 1D nanobeam cavity using direct-write, maskless EBIE. **a** Schematic of the etch process in which a focused electron beam (blue) is scanned along the outer sidewalls of the nanobeam cavity to induce the etch reaction in the presence of water molecules. **b** Photoluminescence (PL) spectra of a 1D photonic cavity before tuning (black), and after the first (red) and second (blue) tuning steps were performed by EBIE. **c** Q-factor of the F0 (circles) and F1 (triangles) modes measured immediately after fabrication (black), and after the first (red), and second (blue) tuning steps

comparison with materials that are known to host quantum emitters in the visible range and have the potential for monolithic integration of emitter – cavity systems. These attributes make them particularly appealing to quantum photonic applications. Cavities fabricated using such materials have been shown to have the following Q-factors: YSO – Q ~3000 at $\lambda = 596$ nm[4], GaN – Q ~5200 at $\lambda = 461$ nm[39], 4H SiC – 6700 at $\lambda = 700$ nm[40] and diamond – Q ~11,000 at $\lambda = 734$ nm[41].

The Q-factors obtained from our hBN cavities are indeed promising for quantum photonic applications. Purcell enhancement of quantum emitters that are located optimally in the cavity field maximum is given by Eq. 1, where $\lambda$ is the cavity resonance wavelength in free space, $n$ is the refractive index at field anti node, and $V$ is the mode volume. Using the ladder cavity, with Q ~2100, a Purcell enhancement of ~110 can be achieved.

$$F_p = \frac{3}{4\pi^2} \left(\frac{\lambda}{n}\right)^3 \frac{Q}{V} \qquad (1)$$

**Tunable photonic cavity**. Next, we demonstrate that the optical modes of the fabricated cavities can be tuned deterministically by EBIE. Tuning is essential to match the resonances to the emission wavelengths of quantum emitters, particularly in the case of hBN where the emission wavelength varies substantially and controlled fabrication of emitters with a given wavelength is yet to be demonstrated. In dielectric cavities, the resonant mode strongly depends on cavity parameters such as the width and thickness. Therefore, numerous methods exist to tune cavity modes, such as thinning via oxidation or RIE, deposition of thin dielectric layers, or gas condensation. In this study, we demonstrate the use of EBIE to controllably tune individual cavities to shorter wavelengths without significant degradation of the Q-factor. As is shown schematically in Fig. 3a, the focused electron beam is guided along the outer sidewalls of a cavity, selectively etching these regions and reducing the width of the beam. In Fig. 3b the PL spectrum of a cavity is shown before tuning (black) with modes F0, F1, and F2 at wavelengths of 745.2, 771.7, and 797.8 nm respectively. A reduced width of the 1D cavity by ~10 nm results in a lower effective refractive index for the optical modes, which resulted in a blue-shift of the cavity resonance by 2.4 nm (red) after the first tuning step. The same process was repeated again to demonstrate further tuning of an additional 4.2 nm (blue). It can be seen (Fig. 3c) that overall the Q-factor of the fundamental mode did not degrade significantly (a change of less than 10%) while that of the first-order mode increased in both steps, and the second-order mode showed a mixed behavior after

tuning. We attribute the different trends in Q-factor change to different positions of the field intensity maxima along the beam, and the fact that the EBIE process used for tuning alters the roughness profile of the sidewall (i.e., the roughness of locally smooth regions and those containing asperities can decrease/ increase, respectively, during tuning). However, the repeatable tuning by EBIE can be seen as a reliable, robust method for tuning of PCC optical modes over a relatively wide spectral range. It does not cause substantial damage to the cavity and does not require subsequent RIE or deposition steps. The EBIE approach is therefore attractive for two reasons. First, being a localized, direct-write technique, it can be applied to individual cavities without affecting the modes or Q-factors of neighboring structures on the same chip. This is very challenging using deposition or RIE thinning techniques as they require additional masking steps and the low refractive index of hBN necessitates the use of suspended structures to realize high Q factors. Second, EBIE does not require any postprocessing such as annealing that is needed to remove FIB-induced damage, or wet-etching used to etch regions processed by laser oxidation. As a result, the EBIE process can be applied iteratively to individual cavities on a single chip until the desired tuning is achieved.

**Creation of quantum emitters**. Finally, we discuss the potential of integrating quantum emitters hosted by hBN with the fabricated PCCs. Coupling of an emitter to an optical mode requires both spatial and spectral overlap between the emitter and the cavity mode. The spatial overlap probability can be increased by deterministically creating single photon emitters in a non-destructive way with high precision. However, unlike the cases of other materials such as diamond and SiC, ion implantation and electron irradiation have not, to date, been shown to be reliable methods for the generation of quantum emitters in hBN. Furthermore, methods relying on local strain field engineering or laser irradiation are not applicable in this case, as they will cause strong deformation of the suspended cavity and severe damage in the material, resulting in reduction of Q-factors. Hence, to create emitters in hBN, we employed an additional annealing step at 850 °C after all the fabrication steps, as this method is known to generate single photon emitters[20] (albeit in random locations). To demonstrate the efficiency of this process, we compare the two regions shown in Fig. 4a – an unprocessed area on the left and a processed area that contains several nanobeams on the right. The corresponding PL map is shown in Fig. 4b. In the analyzed regions, we found a total of 13 single photon emitters, whose position are marked with yellow circles, all of which are located

 5

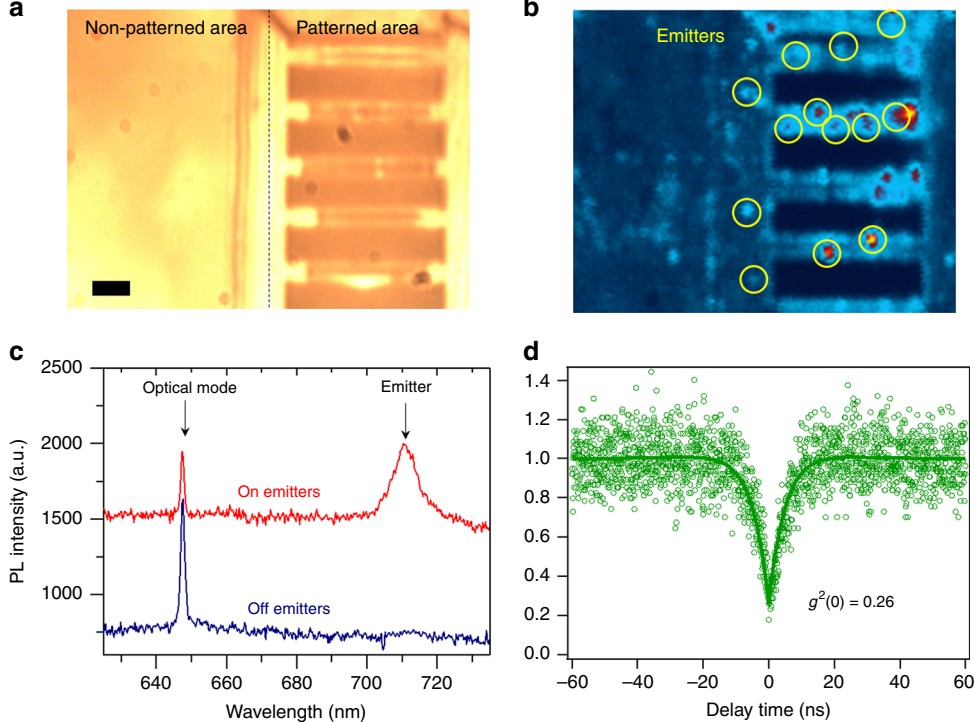

**Fig. 4** Generation of single photon emitters within the hBN cavities. **a** optical microscope image of the analyzed area comparing an unprocessed site (left), with a processed site (right) that contains several 1D nanobeam cavities. The scale bar corresponds to 1 μm. **b** corresponding Photoluminescence map of this region. Positions of quantum emitters are indicated by yellow circles **c** PL spectra from two regions of the same cavity showing an optical mode only (blue) and the combination of an optical mode and an emitter (red). **d** Measured $g^2(\tau)$ curve obtained from this emitter

directly in the fabricated cavities compared with zero emitters found in the unpatterned area. The creation of extra emitters in the cavity-containing regions can be attributed to an increased surface area, creation of dangling bonds, activation of passivated defects, or chemical modification of sites that undergo partial chemical reactions during etching (we note that stable bright emitters were found consistently on the nanobeams only in samples processed by both RIE and EBIE, but not in samples processed only by RIE or only by FIB, irrespective of whether or not the samples were annealed). Bright spots in the confocal map were analyzed spectrally and show clear narrowband zero phonon lines (ZPLs). In several cavities, we clearly observed spatial overlap between cavity modes and emitters, as is shown in Fig. 4c. The top plot (red) includes the ZPL at 710 nm from a color center and the optical mode of the cavity at 647.7 nm. The emitter is within the laser spot, which has a diameter of ~400 nm. Once the laser is spatially detuned from the emitter (bottom, blue curve), only the cavity mode remains, as expected.

To confirm that the peak is indeed a single photon emitter, the emission lines are isolated by spectral filtering and the quantum nature is demonstrated using a Hunbury Brown and Twiss interferometer. The corresponding $g^2(\tau)$ curve with a zero delay time of $g^2(0)=0.26$ (measured without background subtraction) is shown in Fig. 4d, thus classifying it as a single photon source. A further discussion on the characteristics of the observed emitters is given in the Supplementary Note 9.

We note that to demonstrate coupling between emitter and the optical mode, both spatial and spectral matching are required. However, despite the fact that the probability of spatial matching was increased by the processing steps used to fabricate the cavities (Fig. 4b), spectral matching was not observed because the emitters disappeared after an EBIE tuning step. This most likely happened because of the removal of material from the cavity edges during tuning, where the emitter was physically located. At this stage it

has been difficult to find an emitter located in the middle of the dielectric mode of the cavity. Nevertheless, our method of tuning the cavity resonances, along with recent progress in fabrication of the emitters on demand[27,42,43] is promising for eventual realization of coupled monolithic emitter-cavity systems made from hBN.

For hBN to mature into a reliable platform for integrated nanophotonics, an effort into fabrication of membranes with predefined thickness is required, with an emphasis on controlled growth of large-area multilayers. This need is analogous to that which existed in the field of diamond photonics, which has been reinvigorated dramatically when growth of high quality single crystal material became available. We envision such progress will be achieved with BN as well. In addition to the efforts being made toward deterministic growth of BN, further developments in techniques for both deterministic fabrication of quantum emitters in hBN and resonant wavelength tuning will make weak and strong coupling experiments feasible. To realize coupling between emitters and cavities, tuning into resonance will be required. Gas condensation can be used, in principle, for emitters that are in a close proximity (spectrally) to the cavity mode and overlap spatially with the mode. Alternatively, the hBN cavities can be mounted on stretchable substrates and strain tuning may be employed. This would change cavity dimensions homogeneously in one direction without introducing additional scattering centers. Our present demonstration of hBN PCCs is already highly promising for applications in integrated on-chip quantum nanophotonics, optomechanics, cavity QED and quantum sensing experiments. Moreover, integration of hBN cavities with other 2D materials, may yield hybrid heterostructures for studies of light confinement at the nanoscale, and thus position 2D van der Waals systems as a unique platform in the field of integrated nanophotonics. Finally, the demonstrated ability to deterministically fabricate and tune hBN resonators will further expand its

applicability as a hyperbolic material for explorations of light confinement and polaritonics in the mid and near infrared spectral ranges.

## Discussion

In summary, we have demonstrated the fabrication of photonic crystal cavities with Q-factors in excess of 2000 from hBN, a promising van der Waals crystal. We employed both EBIE etching and FIB milling in protocols used to fabricate 1D and 2D PCCs. We also demonstrated a technique to spectrally tune the optical mode of individual cavities using a maskless EBIE method. Finally, we showed that the cavity-patterned regions host a high density of quantum emitters, marking a first step toward monolithic integration of a hBN emitter–cavity system.

## Methods

**Preparation of suspended hBN**. We used trenched silicon substrates in order to maximize the refractive index contrast in z-direction (normal to the plane of 2D hBN layers). The fabrication process from exfoliation to final cavity for the fabrication route via EBL-RIE-EBIE is schematically depicted in Supplementary Note 2. First, high crystalline hBN flakes were mechanically exfoliated from sticky tape onto trenched silicon substrates. Tape residuals were removed by calcination in air for 2 h at 450 °C on a hot plate. Further desorption of contaminants and an increase of adhesion of flakes to the substrate was achieved by annealing in a tube furnace (Lindberg/Blue M) with Argon at 850 °C for 30 min.

**EBIE etching method**. A 15 nm tungsten film was deposited in a homebuilt plasma sputter deposition chamber using a growth rate of ~0.8 Å/s. On top of the tungsten layer, 950 PMMA A3 was spun on to a thickness of ~100 nm. Cavity designs were patterned via conventional EBL at 20 kV, 40 pA, on a commercial Zeiss Supra 55VP SEM with a RAITH E-beam Lithography System. The cavity patterns were transferred into the tungsten film via RIE (homebuilt) using $SF_6$ flown at a rate of 60 sccm and a pressure of 10 mTorr (RF power = 100 watt, self-bias = 200 V). Under these conditions the tungsten film was fully etched in the exposed areas within 20 s. The PMMA was removed in an Acetone bath followed by ashing of residual cross-linked PMMA in Oxygen plasma. Further cleaning was done in Acetone and IPA. EBIE of hBN was done at 1 kV, 15 nA in a water atmosphere of 150 mTorr in a Zeiss Evo LS15. Finally, the tungsten mask was removed using $H_2O_2$ (30%). Although we observe optical modes without further processing steps, the chance of activating/ creating emitters is highly increased after annealing. Therefore, after fabrication the samples were annealed in a tube furnace in Argon, at pressure of 1 Torr, and Argon flow rate of 50 SCCM at 850 °C for 30 min.

**FIB milling**. Cavities fabricated by focused ion beam (FIB) were milled using a $Ga^+$ ion beam at 30 kV, using beam current of 32 nA. Water vapor was delivered to the processing site through a gas injection nozzle in order to suppress charging artefacts during FIB milling. In order to remove damaged areas of hBN and remove implanted Ga ions, the samples were annealed for 2 h in vacuum (1 mTorr) in a tube furnace (Lindberg/ Blue Minimite) at 900 °C.

**Optical characterization**. Optical measurements were performed in a setup as shown in Supplementary Note 10. Broad band emission in hBN and single emitters were excited by a continuous-wave 532 nm laser (Gem 532, Laser Quantum Ltd.). The laser beam passed a polarizer and a half waveplate, and was focused using a high numerical aperture objective lens (NA = 0.9, Nikon). The acquired signal from the sample was collected using the same objective lens, then separated from excitation by a dichroic mirror (BrightLine, Semrock). After the dichroic mirror, further spectral filtering was achieved using a tuneable 20 nm bandpass filter. A flip mirror guided the light either to a Hanburry-Brown and Twiss (HBT) Interferometer or a Spectrometer (Acton SpectraPro, Princeton Instrument Inc.). The HBT setup consists of two avalanche photodiodes (APD, Excelitas Technologies), with a 100 ns delay time induced in one of the APDs connected to a time-correlation card (PicoHarp 300).

**FDTD simulation**. Numerical modeling was performed by using the 3D finite-difference time-domain (FDTD) method (Lumerical Inc). The refractive indices for hBN are $n_x = n_y = 1.84$, $n_z = 1.72$. Two dimensional cavities were designed by introducing line defects in a periodic 2D photonic crystal lattice. One dimensional cavities were designed by modulating the lattice constant.

**Data availability**. The data that support the findings of this study are available from the corresponding author upon request.

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

## Acknowledgements

The authors thank Dr T. Tran for assistance with Raman measurements and Dr C. Elbadawi for the assistance with EBIE. K.W. and T.T. acknowledge support from the Elemental Strategy Initiative conducted by the MEXT, Japan, and JSPS KAKENHI, Grant Number JP15K21722. Financial support from the Australian Research council (via DP140102721, DP180100077, LP170100150), the Asian Office of Aerospace Research and Development grant FA2386-17-1-4064, the Office of Naval Research Global under grant number N62909-18-1-2025 are gratefully acknowledged. I.A. acknowledges the generous support provided by the Alexander von Humboldt Foundation.

## Author contributions

S.K., I.A and M.T conceived the idea and designed experiments. J.E.F. developed the fabrication procedure for EBIE and tuning of cavities. S.K. performed the optical measurement and FDTD simulation. J.C. and M.S. provided cavities fabricated by FIB milling. K.W. and T.T. grew the hBN samples. J.B. assisted with EBIE experiments. D.T. prepared trenched silicon substrate. S.K., J. E. F., I.A., and M.T. co-wrote the manuscript.

## Additional information

**Competing interests:** The authors declare no competing interests.

