## [Peer Review File · Nature Communications]

Reviewers' comments:

Reviewer #1 (Remarks to the Author):

The manuscript from S. Kim et al. reports convincing evidence that layered hexagonal Boron Nitride (hBN) can be nano-fabricated to realize basic nanophotonic components, such as photonic crystal cavities in suspended membranes. The intrinsically low index contrast of hBN is such that it was non-trivial to expect that 2D photonic crystal cavities with sufficiently high quality factors could be realized. The largest Q-factor obtained in this work is on the order of 2000, not remarkably high in absolute terms, but still quite interesting since it occurs in the visible range (around 550 nm wavelength) and in a low-index material. The main novelty of the work is probably the demonstration that a combination of direct e-beam writing and reactive ion etching is effective in nano-patterning the layered membranes, previously suspended through exfoliation on a pre-patterned substrate made of trenches in silicon. The authors also show that a combination of focused ion beam (FIB) and annealing, as an alternative to e-beam+RIE, is actually a very good alternative allowing to achieve the highest reported Q-factors above 2000. Detailed account of the different etching methods is reported in the Supplementary file. Finally, the work is completed by an experimental demonstration that single-emitter type defects can be induced in the hBN films by applying an additional annealing step at 900 degrees Celsius after nanofabrication of the cavities. This approach is shown to yield a sufficient number of (randomly) localized single photon emitters, their quantum nature being probed through antibunched emission by measuring the second-order correlation function from single luminescence spots.

Overall, I appreciated this work, it is well conceived and technically valid, as far as I can judge from careful reading. The steps undertaken to establish hBN as a promising material for nanophotonics at room temperature are well assessed: showing that photonic crystal cavities can be reliably fabricated and their resonances tuned in post-processing are important first steps. However, I am not fully convinced that this manuscript is worth being published in a high impact interdisciplinary journal as Nature Comm., as it stands. A few comments are appended below. If the authors can address the major points raised, and consequently present a suitably revised version, I'd be happy to reconsider. Otherwise, I would suggest to submit to a more specialized journal, possibly focused on materials aspects.

1) The possibility of inducing quantum emitting defects in hBN has already been shown in monolayer hBN, from the same group, in a Nature Nanotech. publication a couple of years ago. Here they go one step further, showing the possibility to fabricate photonic crystal cavities in layered hBN with thickness in the range 200-500 nm, where quantum emitters can also be induced by post-processing of the photonic crystal cavities. However, at this point it might be expected that the authors could show compelling evidence of coupling between a single-defect emitting anti bunched radiation and a mode, possibly with measurement of lifetime change. I think the latter would be a result worth being communicated to the physics community at large, otherwise the main novelty resides in the materials aspect and fabrication techniques applied to this material, which does not represent a sufficient case for publishing in Nature Comm., in my opinion. Once demonstrated the two pieces of the story, i.e. (relatively) high-Q photonic crystal cavity and antibunched emitters, I would rather ask the authors if a further effort is possible, and coupling between the two could be shown convincingly. Then I would be in favor of publication in Nat Comm.

2) There is something that is not totally clear to me in the results shown in fig. 2, or something fundamental I am missing about 1D photonic crystal cavity modes. The authors show multiple resonance of a 1D stack of layered hBN trenches, where successive harmonic modes are reported and spectrally isolated in PL. However, the fundamental mode (F0) is shown to be at shorter wavelength than the first excited (F1), which in turn is at shorter wavelength than the second excited (F2). These results are also obtained from FDTD simulations, as detailed in the supplementary information file. However, this is a bit counterintuitive, I think the authors should

clarify this aspect of the photonic crystal cavity design employed.

3) The authors try to promote this platform for potential impact in integrated photonics. However, it is objectively true that this is not exactly the most deterministic type of material platform, since it relies on exfoliation of layered films with unpredictable thickness, up to 500 nm. It seems that the authors should discuss about the statistics of hBN thickness obtained when exfoliating the material onto the trenched Si substrate, and how this can be turned into a useful technology given that there seems to be no control over this parameter (as, e.g., in a epitaxial growth). In particular, it would be interesting to know if there is any perspective for getting films with deterministic thickness, or if this is not perceived as a limitation for a practical technology to develop. In this reviewer opinion this would be a crucial discussion to establish any such material in comparison to traditional nanostructure semiconductors.

4) A minor curiosity to satisfy; Why did the authors decide to realize L9 cavity as an example of 2D PCC? What is special with this design?

5) In Fig. 3 the authors show results on post-processing tuning of the cavity resonances, in particular the effect that a direct massless e-beam induced etching onto the 1D PCC has on the resonances Q-factor. Here it seems that the Q-factor of the fundamental (F0) mode slightly decreases on increasing tuning step, while the second order mode (F1) slightly increases its Q. Can the authors give an argument for this behavior? From the theoretically calculated mode profiles in Fig. 2c, it does not seem the effect should be that different for the two modes.

Reviewer #2 (Remarks to the Author):

The letter by S. Kim, J. E. Fröch et. al. shows the first results on a photonic crystal cavity made of hBN crystal fabricated from two different methods showing rather good Q factors. The main interesting claim, beside the proof of concept, is the individual and reproducible tunability of these cavities with a rather small loss in the Q factor.

1) Although the authors present scientifically accurate results which could open new perspectives in the community, the overall manuscript guideline is not always clear and a precise discussion on the impact of such discoveries is lacking.

2) The introduction should have a deeper discussion on the advantages of using hBN as a photonic crystal despite the various drawbacks mentioned (low refractive index, technical difficulties...).

3) Although I think efforts have to be made towards clean and controlled etching of vdW materials, I do not agree with the statement of the paragraph, starting line 56, where the authors claim that « nanofabrication protocols for van der Waals layered materials are not yet established » since a wide literature is already accessible on the subject with nice results in electronic transport, nanomechanics and so on. This paragraph needs to be rephrased and referenced to truly describe the actual context.

4) Clean etching is probably one of the main reason leading to variation of the quality factor in these devices as mentioned line 171, but no comments has been made on the quality of the grown hBN crystals. In fact, the 2d materials community faces problems concerning the growth of large, high quality materials, with hBN being one of the harder to grow. Comments on the quality of the crystals used and the possible effects on the device performance of different type of defects would be useful in the discussion by comparing with the various studies on that matter.

Therefore, I think that this manuscript is not matching the broad audience standard of Nature Communication in its current form and needs to be modified prior any submission.

Reviewer #3 (Remarks to the Author):

In Photonic Crystal Cavities in Layered van der Waals crystals, the authors report on techniques for the design and fabrication of tunable photonic crystal cavities (PCCs) in hexagonal boron nitride (hBN) via several techniques. The report of the fabrication of PCCs in van der Waals materials is a novel thing, and may well have implications in creating on-chip devices with PCCs which have the ability to host quantum emitters (QEs), taking full advantage of the QE rather than using existing hybrid coupling modes between QEs and PCCs. This paper will be of interest to others working in the field of nanophotonics and quantum technologies, especially those focusing on developing nanophotonics in novel materials systems (such as van der Waals materials).

My feeling is that as it stands, this work is not broad enough in scope to be published in Nature Communications. The paper consists of two main parts: (1) the design and fabrication of the PCCs in hBN and (2) observations and characterization of QEs in the resultant hBN PCCs. Currently, the two parts seem disconnected, as though they're separate stories. The paper shows that QEs exist in the patterned, suspended hBN, but the details of how this fits in with the PCC size/shape is unclear. Inclusion of the thorough characterization of QEs in the various hBN PCCs and a more fleshed-out means of successfully incorporating these van der Waals PCC fabrication techniques with the QEs would help connect these two parts and strengthen the claim that these hBN PCCs will find use as on-chip photonic resonators with embedded QEs.

The manuscript is clearly written. It could be shortened to keep the focus on the fabrication of the PCCs, but doing so would significantly limit the interested audience for this paper to a very focused set of researchers. The authors have done themselves justice without overselling their claims regarding the fabrication of PCCs in hBN, though I feel as though the capability for QE integration with the PCCs is not as strong as the authors currently make it out to be.

Some additional comments:

1. Line 1 - The current title of the paper referring to van der Waals crystals in general (and not hBN in particular) necessitates the inclusion of more context regarding the role played by other van der Waals materials such as the transition metal dichalcogenides (whose QEs are strain-induced and likely different from those observed in hBN). The role of these other van der Waals materials as PCCs integrated with QEs should be discussed if the title is to stay as is.
2. Line 53 – Consider including Exarhos et al., ACS Nano, 2017, 11 (3), pp 3328–3336 as a reference here. This paper is directly relevant as it focuses on the various properties of hBN QEs in suspended samples and supports the observations of substrate effects.
3. Line 62 – What wavelength does the refractive index correspond to?

4. Line 76 – varies \propto vary
5. Line 161 – include references for the sensing applications for S0
6. Line 166 – Fig. 2d: Is the claim that roughness is primarily responsible for the significant variation in Q for F0? The justification of higher Q for longer wavelength is fine, but the significant jump from Q at 725 nm to longer wavelengths makes me wonder what the variation in Q is for a particular resonance wavelength.
7. Line 170 – The theoretical Q calculated does not correspond to the dimensions of the 1D cavity that was measured. How does the theoretical Q change for the particular cavity dimensions measured experimentally?
8. Some discussion on the variation of Q with nanobeam thickness may be warranted. Fig. S6 shows the modelling where Q increases with increasing thickness, so it would be worth considering the limiting thickness of these PCCs in hBN. It's unlikely they'll be successfully realized in few-layer hBN.
9. When exfoliating onto the trenches, are the films thick enough to avoid any wrinkling or sagging due to being suspended?
10. Line 182 – should read 900 oC
11. Line 196 – Q's for PCCs in hBN are on par, but fall consistently below other bulk dielectrics. Furthermore, the optimal Q resonance wavelength varies for fabrication technique. I think there could be some discussion here about possible variations in Q over resonance wavelengths in the VIS range for different PCC fabrication techniques.
12. Line 225 – “reduced width” – By how much?
13. Line 227 – What is the full tuning range possible with this etching techniques? One might conceivably need to tune over hundreds of nm to match the cavity resonance with the QE zero phonon line. This is particularly relevant as the motivation for developing the means of tuning the cavity resonance is to match the QE ZPL (lines 214-215). Something akin to Fig. S6e might be useful.
14. Figure 3c – Worth plotting QF^2 as well? It appears to increase with tuning # in Fig. 3b. Any

reason for why QF1 and QF2 might behave oppositely from QF0?

15. Line 260 – specify annealing in Ar. It's stated that EBIE is preferable because annealing is unnecessary, but could RIE and annealing cover both the Q-factor increase and the QE formation simultaneously? What are the annealing conditions to heal the sample after RIE?

16. Lines 265-269 – Fig. 4c alone is not convincing to me. The distance of the QEs on the nanobeams from the edges doesn't look remarkably different from the distance to the edge on the unpatterned region. Correlations between spectral shape and ZPL wavelength, optical dipole orientations, and photon emission dynamics in the patterned and unpatterned regions would make a stronger case for the conclusions here. What about the characteristics of the QEs in cavities of other sizes?

17. Line 271 – Show the additional ZPL data (SI would suffice).

18. Line 283 – Do other QEs survive the cavity tuning via EBIE (such as those note on the particular nanobeam being etched)?

19. Line 297 – caption refers to panel (d) twice. Change the final designation to panel (e).

20. Line 315 – Discussion of the implication of hBN nanophotonics in mid and near infrared spectral ranges should be expanded and incorporated more into the main focus of the paper, which is on resonant wavelengths in the VIS part of the spectrum (in order to match QE ZPLs).

21. Lines 323, 324, 346 – degrees C notation is not rendering properly

Reviewer #1:

Overall, I appreciated this work, it is well conceived and technically valid, as far as I can judge from careful reading. The steps undertaken to establish hBN as a promising material for nanophotonics at room temperature are well assessed: showing that photonic crystal cavities can be reliably fabricated and their resonances tuned in post-processing are important first steps.

However, I am not fully convinced that this manuscript is worth being published in a high impact interdisciplinary journal as Nature Comm., as it stands. A few comments are appended below. If the authors can address the major points raised, and consequently present a suitably revised version, I'd be happy to reconsider. Otherwise, I would suggest to submit to a more specialized journal, possibly focused on materials aspects.

We thank the reviewer for the positive feedback on our work, and recognition of the challenges to fabricate high Q devices from low index layered materials. Below, we addressed the reviewer's comments in detail.

1) The possibility of inducing quantum emitting defects in hBN has already been shown in monolayer hBN, from the same group, in a Nature Nanotech. publication a couple of years ago. Here they go one step further, showing the possibility to fabricate photonic crystal cavities in layered hBN with thickness in the range 200-500 nm, where quantum emitters can also be induced by post-processing of the photonic crystal cavities. However, at this point it might be expected that the authors could show compelling evidence of coupling between a single-defect emitting anti bunched radiation and a mode, possibly with measurement of lifetime change. I think the latter would be a result worth being communicated to the physics community at large, otherwise the main novelty resides in the materials aspect and fabrication techniques applied to this material, which does not represent a sufficient case for publishing in Nature Comm., in my opinion. Once demonstrated the two pieces of the story, i.e.(relatively) high-Q photonic crystal cavity and antibunched emitters, I would rather ask the authors if a further effort is possible, and coupling between the two could be shown convincingly. Then I would be in favour of publication in Nat Comm.

We acknowledge this request from the reviewer, and indeed coupling of emitters to cavities is one of our long-term goals. There are a few experimental challenges that prevent us from convincingly demonstrating it at this stage. In most semiconductors (GaN, GaAs), spectral matching is achieved by 'thermal' and 'electrical' (by Stark effect) tuning of the QD to the cavity mode. However, for SPEs in wide bandgap materials, such as diamond and hBN, the difficulty lies in the fact that they are robust to thermal and electrical tuning methods (similarly, it works with NV centers, but only for several GHz). Therefore, in the case of diamond, coupling was realized by 'gas condensation' to tune the cavity mode instead of the ZPL, which is a much harder process. We accomplished the first step towards spectral matching by tuning the cavity mode through deterministic narrowing of the 1D photonic crystal nanobeam as described in our manuscript. This, however, can't be used to achieve the final fine-tuning that is needed during a PL measurement, and must be combined with the gas condensation method. Yet, finding hBN SPEs close to cavity modes is challenging due to the wide energy distribution of ZPLs in hBN samples that can be grown at present – hence, further work on the topic of engineering emitters with pre-defined ZPL energy is required, and we believe that our results will motivate the rapidly growing BN community to expedite these lines of research. Our results provide the first evidence for high-Q cavity

nanofabrication and we are certain it will be broadly appealing to not only BN, but also to the 2D materials and photonics communities.

We added a short description on the most promising pathway to achieving Purcell enhancement at the end of our manuscript:

“To realize coupling between the emitters and the cavity, tuning into resonance will be required. Gas condensation can be used, in principle, for emitters that are in a close proximity (spectrally) to the cavity mode and overlap spatially with the mode. Alternatively, the hBN cavities can be mounted on stretchable substrates and strain tuning may be employed.”

2) There is something that is not totally clear to me in the results shown in fig. 2, or something fundamental I am missing about 1D photonic crystal cavity modes. The authors show multiple resonance of a 1D stack of layered hBN trenches, where successive harmonic modes are reported and spectrally isolated in PL. However, the fundamental mode (F0) is shown to be at shorter wavelength than the first excited (F1), which in turn is at shorter wavelength than the second excited (F2). These results are also obtained from FDTD simulations, as detailed in the supplementary information file. However, this is a bit counterintuitive, I think the authors should clarify this aspect of the photonic crystal cavity design employed.

The frequencies of the fundamental mode and higher order modes are determined by the photonic bands used for the cavity design. In our case, the lowest energy photonic band forms a photonic well as shown in Figure R1. The fundamental mode has the highest energy, and is therefore located at the shortest wavelength. We added this explanation and a reference to the

supporting information section 5.

Figure R1. Photonic well. [B.-H Ahn et. al., Optics Express, 18, 6, 2010]

3) The authors try to promote this platform for potential impact in integrated photonics. However, it is objectively true that this is not exactly the most deterministic type of material platform, since it relies on exfoliation of layered films with unpredictable thickness, up to 500 nm. It seems that the authors should discuss about the statistics of hBN thickness obtained when exfoliating the material onto the trenched Si substrate, and how this can be turned into a useful technology given that there seems to be no control over this parameter (as, e.g., in an epitaxial growth). In particular, it would be interesting to know if there is any perspective for getting films with deterministic thickness, or if this is not perceived as a limitation for a practical technology to develop. In this reviewer opinion this would be a crucial discussion to establish any such material in comparison to traditional nanostructure semiconductors.

Control over material thickness is indeed important for practical devices and it is required for integration of this material into photonic chips. First, to address the reviewer’s comment on the statistics of hBN thickness, we added SEM images of 4 hBN flakes to the Supplementary

Information (Section 1) to show the thickness variation. The thickness of flakes after tape exfoliation typically varies between ~ 200 and 600 nm (see Figure R2 below).

Figure R2. SEM images of hBN flakes directly after exfoliation.

Next, regarding the reviewer’s concern about uncertainty in the thicknesses of hBN flakes, we note that several groups are working on controllable wafer-scale CVD growth of hBN, as has been demonstrated in recent papers [Jang, A-Rang, et al. *Nano letters* 16.5 (2016): 3360-3366., Behura, Sanjay, et al. *ACS nano* 11.5 (2017): 4985-4994]. In our work, we exfoliated hBN flakes onto a trenched substrate because it is a convenient means to create a freestanding geometry, which otherwise requires a well-defined, high-selectivity undercut process that is presently not available. There is, however, no reason to doubt that hBN fabrication processes accurate to a monolayer will eventually be developed, as will undercut techniques. In fact, the ability to manipulate layered materials as we have done here will likely stimulate interest in and accelerate the development of such techniques because proof-of-concept studies such as ours are not delayed by the rate of development of conventional nanofabrication techniques.

For sake of perspective, it is important to note that despite the rapid rate of progress in the field of 2D materials, engineering of device-grade BN samples is in its infancy compared to other systems such as GaAs or even diamond. In fact, the original work on diamond photonics required cumbersome processing and was initially applicable only to polycrystalline materials [Wang, Chiou-Fu, et al. *APL* 91 (2007) 201112.]. However, within less than a decade, high quality membranes became available and enabled true “integrated photonics”. At this stage, we see no fundamental barrier preventing BN from becoming the next platform for nanophotonics. We expanded this discussion by adding the following paragraph to the conclusion of the main manuscript:

“for BN to mature into a reliable platform for integrated nanophotonics, an effort into fabrication of membranes with pre-defined thickness is required, with an emphasis on controlled growth of large-area multilayers. This need is analogous to that which existed in the field of diamond photonics, which has been reinvigorated dramatically when growth of high quality single crystal material became available. We envision such progress will be achieved with BN as well”.

4) A minor curiosity to satisfy; Why did the authors decide to realize L9 cavity as an example of 2D PCC? What is special with this design.

Typically, L3 cavities are most commonly used in higher refractive index materials (like GaAs/Silicon/Diamond). In this work with low refractive index material, we used longer cavity (L9) to get higher Q-factors. However, as the number of missing holes increases, the mode volume increases as well, which would in turn reduce the Purcell enhancement of the device, thus a certain trade-off has to be met.

5) In Fig. 3 the authors show results on post-processing tuning of the cavity resonances, in particular the effect that a direct maskless e-beam induced etching onto the 1D PCC has on the resonances Q-factor. Here it seems that the Q-factor of the fundamental (F0) mode slightly decreases on increasing tuning step, while the second order mode (F1) slightly increases its Q. Can the authors give an argument for this behavior? From the theoretically calculated mode profiles in Fig. 2c, it does not seem the effect should be that different for the two modes.

This is not unexpected because of the random distribution of asperities responsible for the sidewall roughness observed after cavity fabrication (i.e. before tuning), the fact that the EBIE process used for tuning alters these asperities, and the fact that the intensity maxima of each mode are distributed over different regions of the beam. As is shown in Fig. 2c of the manuscript, the intensity maximum of the F0 mode is located directly in the middle of the nanobeam, while the intensity maximum of F1 has a minimum at the nanobeam center. The area processed by EBIE to achieve tuning is a sidewall, and the EBIE process alters the roughness profile of the sidewall – in particular, the roughness of locally smooth regions is increased, and local asperities that protrude out from the beam are smoothed-out. Hence, the net effect on Q-factor depends on the distribution of the asperities relative to the mode maxima. We added an explanatory comment to the manuscript, and we additionally included the data for the F2 mode to Figure 3c.

Reviewer #2:

The letter by S. Kim, J. E. Fröch et. al. shows the first results on a photonic crystal cavity made of hBN crystal fabricated from two different methods showing rather good Q factors. The main interesting claim, beside the proof of concept, is the individual and reproducible tunability of these cavities with a rather small loss in the Q factor.

1) Although the authors present scientifically accurate results which could open new perspectives in the community, the overall manuscript guideline is not always clear and a precise discussion on the impact of such discoveries is lacking.

We thank the reviewer for acknowledging that our results can indeed open new perspectives in the community. To improve the manuscript clarity, we followed the reviewer's suggestions.

2) The introduction should have a deeper discussion on the advantages of using hBN as a photonic crystal despite the various drawbacks mentioned (low refractive index, technical difficulties...).

The main motivation to fabricate hBN cavities is to achieve a monolithic platform for integration with its exceptionally promising quantum emitters, as is now clarified in the introduction.

Following the reviewer's comment, we added another paragraph motivating BN nanophotonics in the introduction:

“... hBN is naturally hyperbolic and exhibits volume-confined phonon polariton modes. The phonon polaritons open up opportunities to study light matter interaction at deep subwavelength regime and advance infrared and terahertz nanophotonics. It is therefore important to engineer nanostructures that enable control over a broad spectral range that accommodates both visible on-chip nanophotonics and integrated polaritonics. Finally, the layered nature of hBN allows flakes to be easily relocated and combined with other platforms. This, combined with its inherent chemical inertness, makes hBN resilient to a broad range of environments, including liquids, and hence applications such as sensing where the relatively low refractive index of hBN is beneficial as it leads to large evanescent fields that enhance sensing efficiency.”

3) Although I think efforts have to be made towards clean and controlled etching of vdW materials, I do not agree with the statement of the paragraph, starting line 56, where the authors claim that « nanofabrication protocols for van der Waals layered materials are not yet established » since a wide literature is already accessible on the subject with nice results in electronic transport, nanomechanics and so on. This paragraph needs to be rephrased and referenced to truly describe the actual context.

We agree that simple geometries like rods and cones have been demonstrated from hBN. Yet, functional high quality photonic devices that function in the visible and near infrared spectral range (and thus have low error tolerances) have not been demonstrated using any vdW crystal. We rephrased this paragraph accordingly in the manuscript:

“In contrast to mature semiconductors, nanofabrication protocols for making nanoscale, high quality optical devices in the visible range using van der Waals layered materials are not yet fully developed, and the associated challenges are not fully understood.”

4) Clean etching is probably one of the main reason leading to variation of the quality factor in these devices as mentioned line 171, but no comment has been made on the quality of the grown hBN crystals. In fact, the 2d materials community faces problems concerning the growth of large, high quality materials, with hBN being one of the harder to grow. Comments on the quality of the crystals used and the possible effects on the device performance of different type of defects would be useful in the discussion by comparing with the various studies on that matter.

We agree about the immature state of current nanofabrication/processing techniques, but as noted above, we believe these issues are technical in nature, surmountable, and they will be resolved. In our work, we exfoliated hBN films from bulk hBN crystals grown by a high temperature, high pressure method that has been described in detail elsewhere [Taniguchi, T., and K. Watanabe. *Journal of crystal growth* 303.2 (2007): 525-529.]. These hBN crystals are state-of-the-art – they have favorable PL properties, a relatively low concentration of carbon and oxygen impurities, and from the optical images in the Supplementary Information they appear transparent and smooth. We added this description and the reference to the main text in line 71. Moreover, several groups are currently making progress with large area hBN growth [Jang, A-Rang, et al. *Nano letters* 16.5 (2016): 3360-3366., Behura, Sanjay, et al. *ACS nano* 11.5 (2017): 4985-4994.], and as is mentioned above, a brief discussion of this topic was added to the manuscript.

Reviewer #3:

In Photonic Crystal Cavities in Layered van der Waals crystals, the authors report on techniques for the design and fabrication of tunable photonic crystal cavities (PCCs) in hexagonal boron nitride (hBN) via several techniques. The report of the fabrication of PCCs in van der Waals materials is a novel thing, and may well have implications in creating on-chip devices with PCCs which have the ability to host quantum emitters (QEs), taking full advantage of the QE rather than using existing hybrid coupling modes between QEs and PCCs. This paper will be of interest to others working in the field of nanophotonics and quantum technologies, especially those focusing on developing nanophotonics in novel materials systems (such as van der Waals materials).

My feeling is that as it stands, this work is not broad enough in scope to be published in Nature Communications.

We thank the reviewer for the detailed analysis, and the comment that the paper will be of interest to others working on nanophotonics, quantum technologies and layered materials. This, in our mind, highlights why the paper is in fact suitable for nature communications. We have implemented the recommendations below and further improved the clarity of our work.

The paper consists of two main parts: (1) the design and fabrication of the PCCs in hBN and (2) observations and characterization of QEs in the resultant hBN PCCs. Currently, the two parts seem disconnected, as though they're separate stories. The paper shows that QEs exist in the patterned, suspended hBN, but the details of how this fits in with the PCC size/shape is unclear.

As the reviewer commented, our work can be divided into two parts. Part (1) shows the first demonstration of hBN PCCs. They can be widely deployed in the fields of 'quantum photonics', 'polaritons' and 'lasers', etc. Part (2) shows suitability of the processed material for applications in integrated quantum photonics – our primary motivation for fabricating these cavities. We have added several connecting sentences to clarify the storyline in the paper.

Also, regarding the reviewer's comments on PCC size/shape, we have expanded our discussion in the manuscript to explain why we fabricated different size PCCs – specifically, ZPLs of QEs in hBN samples that are presently available are observed over a range spanning 600 to 750 (e.g. see Figure R4 below). We therefore demonstrated that the PCCs can be both designed and fabricated to have optical modes in this spectral range.

Figure R4. **a**, Spectra of emitters from the patterned area. **b**, **a** PL map of the flake with colored circles showing positions of emitters. **c**, Fluorescence intensity as a function of time demonstrating the photostability of a hBN SPE1. **d**, Lifetime measurement of SPE1 showing an excited-state lifetime corresponding to 3.8 ns. **e**. Antibunching curve with $g^2(0)$ with 0.18.

Inclusion of the thorough characterization of QEs in the various hBN PCCs and a more fleshed-out means of successfully incorporating these van der Waals PCC fabrication techniques with the QEs would help connect these two parts and strengthen the claim that these hBN PCCs will find use as on-chip photonic resonators with embedded QEs.

Spectra from eight QEs are now included in the Supplementary Information and shown in Figure R4. Here, we used a different flake of hBN from the one used for Figure 4 to highlight the

fact that emitters are observed in all of our flakes after PCC fabrication. We also included a detailed analysis of SPE1, showing photostability (532 cw laser, 30 uW), lifetime, and antibunching data.

The manuscript is clearly written. It could be shortened to keep the focus on the fabrication of the PCCs, but doing so would significantly limit the interested audience for this paper to a very focused set of researchers. The authors have done themselves justice without overselling their claims regarding the fabrication of PCCs in hBN, though I feel as though the capability for QE integration with the PCCs is not as strong as the authors currently make it out to be.

While we agree that our approach for making QEs is not entirely deterministic so far, we have demonstrated that numerous quantum emitters are generated around the PCCs using a simple, minimally-invasive method. At this stage, our work lays the foundation for future work on fully deterministic incorporation of emitters in hBN photonic crystal cavities. We also note that both our group and a multitude of others are working actively on engineering of emitters with pre-defined locations and ZPL energies, and there is no reason to believe that these challenges will not be solved in the near future.

Some additional comments:

1. Line 1 - The current title of the paper referring to van der Waals crystals in general (and not hBN in particular) necessitates the inclusion of more context regarding the role played by other van der Waals materials such as the transition metal dichalcogenides (whose QEs are strain-induced and likely different from those observed in hBN). The role of these other van der Waals materials as PCCs integrated with QEs should be discussed if the title is to stay as is.

To make the title more precise and appropriate, we changed it to “Photonic Crystal Cavities from Hexagonal Boron Nitride”.

2. Line 53 – Consider including Exarhos et al., ACS Nano, 2017, 11 (3), pp 3328–3336 as a reference here. This paper is directly relevant as it focuses on the various properties of hBN QEs in suspended samples and supports the observations of substrate effects.

The reference was added.

3. Line 62 – What wavelength does the refractive index correspond to?

The refractive index corresponds to a wavelength of 600 nm. This information was included in the manuscript.

4. Line 76 – varies \diamond vary

This comment has been reflected in the manuscript.

5. Line 161 – include references for the sensing applications for S0

The reference was included.

6. Line 166 – Fig. 2d: Is the claim that roughness is primarily responsible for the significant variation in Q for F0? The justification of higher Q for longer wavelength is fine, but the significant jump from Q at 725 nm to longer wavelengths makes me wonder what the variation in Q is for a particular resonance wavelength.

The roughness of the sidewalls is one of the major factors that affect the Q-factor. Other factors are variations in thickness, and different air hole size from sample to sample due to fabrication errors. Therefore, a certain amount of Q-factor variation exists. The apparent jump in Figure 2d noted by the referee is not physically significant, and caused merely by scatter in our dataset.

7. Line 170 – The theoretical Q calculated does not correspond to the dimensions of the 1D cavity that was measured. How does the theoretical Q change for the particular cavity dimensions measured experimentally?

We included a new theoretical Q-factor using dimensions given for Figure 2a, which are $a=250$ nm, $w=750$ nm, $h_x=150$ nm, $h_y=550$ nm and $t=280$ nm. The theoretical Q-factor is 9,000 with a resonant wavelength of 662.8 nm. The manuscript was modified accordingly.

8. Some discussion on the variation of Q with nanobeam thickness may be warranted. Fig. S6 shows the modelling where Q increases with increasing thickness, so it would be worth considering the limiting thickness of these PCCs in hBN. It's unlikely they'll be successfully realized in few-layer hBN.

Yes. As the reviewer pointed out, it is indeed impossible to achieve photonic resonators with few-layered hBN. This is because all photonic resonators require a certain slab thickness to confine light. Therefore, in our experiment, we chose hBN thicknesses larger than 200 nm so as to make reasonably high-Q cavities. As shown in Figure R2, most of the flakes after exfoliation have sufficient thickness, so there is no significant limiting factor in the fabrication process. To address this point we expanded the associated discussion in the manuscript.

Theoretically, there is an optimum thickness for the slab rather than a limiting thickness. An optimum value exists because both extremes – too thin and too thick relative to the wavelength – cannot confine the light. To show this behavior, we modified Figure S6 to include a wider range of thickness variation as shown in Figure R5. The optimum thickness in this case is approximately 500 nm.

Figure R5. Q-factor and wavelength versus nanobeam thickness.

9. When exfoliating onto the trenches, are the films thick enough to avoid any wrinkling or sagging due to being suspended?

Yes, the hBN films are typically strong enough, thus wrinkling or sagging does not occur when suspended. To emphasize this, we added SEM images to the SI (also shown in Figure R1) of several suspended hBN films that also show a distribution of film thickness from ~200 to 600 nm.

10. Line 182 – should read 900 C

This comment has been reflected in the manuscript.

11. Line 196 – Q's for PCCs in hBN are on par, but fall consistently below other bulk dielectrics. Furthermore, the optimal Q resonance wavelength varies for fabrication technique. I think there could be some discussion here about possible variations in Q over resonance wavelengths in the VIS range for different PCC fabrication techniques.

Q-factors for hBN PCCs in this work are slightly lower than other systems mentioned in the manuscript, but still on the same order of magnitude. We revised the text and substituted “on par with, yet lower”.

Regarding reviewer's comment on Q-variation for different PCC fabrication techniques, it is true that our samples show differences – in FIB samples and RIE+EBIE samples the highest Q-

factors were 2,100 and 1,700, respectively – but we don't believe these differences are fundamental, and the Q factors will be improved over the coming years as the methods are optimised.

Each method does, however, have some clear pros and cons though, and we therefore clarified and expanded our discussion of these. FIB milling can produce smoother sidewall as shown in Figure 2e, but a subsequent annealing step is necessary to remove crystal damage and observe optical modes. On the other hand, RIE+EBIE yields rougher sidewalls, but we can measure optical modes even without a post-annealing step, which is particularly appealing for iterative processes such as tuning and for processing of device structures that cannot withstand high temperature annealing treatments.

12. Line 225 – “reduced width” – By how much?

The cavity width was reduced by approximately 10 nm.

13. Line 227 – What is the full tuning range possible with this etching techniques? One might conceivably need to tune over hundreds of nm to match the cavity resonance with the QE zero phonon line. This is particularly relevant as the motivation for developing the means of tuning the cavity resonance is to match the QE ZPL (lines 214-215). Something akin to Fig. S6e might be useful.

The tuning range of a given cavity is on the order of tens of nanometers. Tuning over hundreds of nm is extremely ambitious and challenging for all existing techniques including temperature tuning, electrical tuning, gas condensation, dielectric deposition, liquid infiltration and physical etching of the cavity. However, tunability over tens of nanometers is adequate even in hBN, despite the broad distribution of ZPL energies in samples that are available at present – for example, we have already found an emitter with a ZPL within 15 nm of cavity modes, as shown in Figure R6 (the emitter was, however, destroyed in an attempted EBIE tuning step, likely because it was located at/near the surface).

Figure R6. **a**, PL spectrum showing ZPL of emitter and two optical modes. Inset shows confocal PL map. **b**, PL spectrum with two optical modes.

14. Figure 3c – Worth plotting QF2 as well? It appears to increase with tuning # in Fig. 3b. Any reason for why QF1 and QF2 might behave oppositely from QF0?

QF2 was added to the figure. We attribute the different trend in Q – factor to differences in spatial positions of the corresponding optical modes, as we explained above (point 3 of reviewer 1).

15. Line 260 – specify annealing in Ar. It's stated that EBIE is preferable because annealing is unnecessary, but could RIE and annealing cover both the Q-factor increase and the QE formation simultaneously? What are the annealing conditions to heal the sample after RIE?

We clarified the annealing process: To create emitters, the samples were annealed in a tube furnace in Argon, at pressure of 1 Torr, with the flow rate of 50 SCCM at 850°C for 30 minutes. Annealing conditions for cavities fabricated by FIB were 900°C for 2 hours in vacuum (1 mTorr).

We also clarified this: Stable bright emitters were found consistently on the nanobeams only in samples processed by both RIE and EBIE, but not in samples processed only by RIE or only by FIB. Furthermore, the EBIE step was needed to improve the sidewall slope (see SI, Figure S4).

16. Lines 265-269 – Fig. 4c alone is not convincing to me. The distance of the QEs on the nanobeams from the edges doesn't look remarkably different from the distance to the edge on the unpatterned region. Correlations between spectral shape and ZPL wavelength, optical dipole orientations, and photon emission dynamics in the patterned and unpatterned regions would make a stronger case for the conclusions here. What about the characteristics of the QEs in cavities of other sizes?

We agree that the three emitters that were categorized, conservatively/pessimistically, as being in the 'un-patterned/processed' region should in fact be classified as being within the 'patterned' region. It is therefore more correct to say that we have '0' emitters in the un-patterned region and 13 emitters in the patterned region (this is also consistent with other cavities that we investigated – in these high purity hBN flakes, there are no optically active emitters in the unprocessed sample regions, as shown in Figure R4, and it is the processing itself that generates/activates emitters). We removed Figure 4c and corrected the manuscript accordingly. We also added data that characterizes the emitters – including the PL spectra, photo-stability curve and lifetime measurement shown in Figure R4. The photophysical properties of the emitters are not atypical in any way relative to those published in prior papers. Here we added more data for additional three emitters found in patterned area.

Figure R7. Lifetime measurement and photostability measurement from Emitter1, Emitter2 and Emitter3.

17. Line 271 – Show the additional ZPL data (SI would suffice).

We included additional ZPL data in the SI as shown in Figure R4.

18. Line 283 – Do other QEs survive the cavity tuning via EBIE (such as those note on the particular nanobeam being etched)?

Quantum emitters on beams which are not on the particular beam that is being edited seem to not survive the EBIE process. As is mentioned in the manuscript, the emitters are likely to be created/ activated by a combination of RIE and EBIE. Emitters that occur naturally (or will eventually be engineered) within the hBN stack are expected to survive the tuning as both RIE and EBI by a low energy electron beam affect only the top few monolayers of the sample.

19. Line 297 – caption refers to panel (d) twice. Change the final designation to panel (e).

This comment has been reflected in the manuscript.

20. Line 315 – Discussion of the implication of hBN nanophotonics in mid and near infrared spectral ranges should be expanded and incorporated more into the main focus of the paper, which is on resonant wavelengths in the VIS part of the spectrum (in order to match QE ZPLs).

21. Lines 323, 324, 346 – degrees C notation is not rendering properly

This comment has been reflected in the manuscript.

REVIEWERS' COMMENTS:

Reviewer #1 (Remarks to the Author):

The authors have satisfactorily answered, with a detailed report, the many points raised from all of the reviewers. Despite not addressing the main point I was mentioning in my report, namely to show a conclusive evidence of coupling between defect emission and cavity mode, I am persuaded by the authors' argument that such an experiment would require to find emitters that are quite close to the cavity resonances, which is totally relying on chance. Overall, it is a bit on the edge, but the manuscript deserves being published in Nat Comm, in my opinion.

Reviewer #2 (Remarks to the Author):

The authors have well answered the question raised. I believe that the modifications brought to the paper have greatly enhanced the reading of the manuscript and I recommend the publication in Nature Communication.

Reviewer #3 (Remarks to the Author):

I appreciate the thoroughness with which the authors addressed my (as well as the other reviewers') questions and comments. I particularly appreciate the inclusion of Fig. S7, particularly panel (a) and also Fig. R7, which go some way to showing the variation in QE characteristics observed in hBN. I do, however, wonder if the emphasis placed on QE characterization is warranted here, particularly in light of the authors' comments that the observed emitters in Fig. 4 are likely near-surface QEs "created/activated by a combination of RIE and EBIE," whereas the cavity tuning is intended for the naturally occurring or engineered QEs which lie deeper in the hBN stack (in response to my question #18). I feel as though the manuscript would be stronger in keeping the focus on the impressive nanofabrication feat the authors have accomplished. Discussion on spectral matching can be in relation to non-RIE/EBIE-generated QEs and less emphasis should be placed on the QEs that are unable to survive the cavity tuning process.

Ideally, of course, I would like to see evidence of cavity tuning to match an QE in a case where the QE survives the tuning process. Such an observation would really tie together the two aspects of this manuscript. While I agree that the cavity tuning technique is promising for spectral matching between the cavity and a specific QE, I feel that without this key piece of data, the manuscript does not have the broad scope necessary for publication in Nature Communications.

Two additional notes regarding the revised text:

1. line 227: REF?
2. line 283: Strain-field engineering of the cavity is discounted because of a resultant Q-factor reduction, but in lines 337-338, strain tuning is discussed as a means of achieving effective coupling between QEs and the cavity. It stands to reason that a compromise could be reached between coupling efficiency and cavity Q-factor, but some discussion on this to tie these comments together is warranted.

Reviewer #3 (Remarks to the Author):

I appreciate the thoroughness with which the authors addressed my (as well as the other reviewers') questions and comments. I particularly appreciate the inclusion of Fig. S7, particularly panel (a) and also Fig. R7, which go some way to showing the variation in QE characteristics observed in hBN. I do, however, wonder if the emphasis placed on QE characterization is warranted here, particularly in light of the authors' comments that the observed emitters in Fig. 4 are likely near-surface QEs "created/activated by a combination of RIE and EBIE," whereas the cavity tuning is intended for the naturally occurring or engineered QEs which lie deeper in the hBN stack (in response to my question #18). I feel as though the manuscript would be stronger in keeping the focus on the impressive nanofabrication feat the authors have accomplished. Discussion on spectral matching can be in relation to non-RIE/EBIE-generated QEs and less emphasis should be placed on the QEs that are unable to survive the cavity tuning process.

We thank the Reviewer for acknowledging our effort in the revised version. We believe the creation and characterization of quantum emitters in the fabricated devices is warranted as it offers a route towards realization of cavity coupling experiments. While further investigation is required to protect pre-existing QEs during EBIE tuning process, EBIE tuning can have advantages for tuning over longer wavelengths. This can be desirable in optomechanics and phonon-polaritons applications.

Ideally, of course, I would like to see evidence of cavity tuning to match an QE in a case where the QE survives the tuning process. Such an observation would really tie together the two aspects of this manuscript. While I agree that the cavity tuning technique is promising for spectral matching between the cavity and a specific QE, I feel that without this key piece of data, the manuscript does not have the broad scope necessary for publication in Nature Communications.

While we agree that it is important to achieve coupling, this goal is difficult to realize at this stage. We believe the first demonstration of high-Q photonic device from van der Waals material, that includes quantum emitters and means to tune the cavity is a sufficient novelty for publication in Nature Communications, as agreed by reviewer #1, #2. We are certain the work will be followed by many groups to expedite the realization of coupling.

Two additional notes regarding the revised text:

1. line 227: REF?

We added the missing reference.

2. line 283: Strain-field engineering of the cavity is discounted because of a resultant Q-factor reduction, but in lines 337-338, strain tuning is discussed as a means of achieving effective coupling between QEs and the cavity. It stands to reason that a compromise could be reached between coupling efficiency and cavity Q-factor, but some discussion on this to tie these comments together is warranted.

We clarified this discussion as per below. Our original statement indicated that strain activation of emitters is still under debate, but even if works, only possible by positioning the material on a pillar – which will destroy the cavity. The tuning by strain is done by homogenous stretching from two opposite sides, that will result in a minimal deformations

*“... Furthermore, methods relying on **local** strain field engineering or laser irradiation are not applicable in this case, as they will cause **strong** deformation of the suspended cavity and severe damage in the material...”*

“...Alternatively, the hBN cavities can be mounted on stretchable substrates and strain tuning may be employed.

This would change cavity dimensions homogeneously in one direction without introducing additional scattering centers...”